# cGAMP Promotes Germinal Center Formation and Production of IgA in Nasal-Associated Lymphoid Tissue

**DOI:** 10.3390/medsci5040035

**Published:** 2017-12-18

**Authors:** Hiromi Takaki, Ken Takashima, Hiroyuki Oshiumi, Akira Ainai, Tadaki Suzuki, Hideki Hasegawa, Misako Matsumoto, Tsukasa Seya

**Affiliations:** 1Department of Microbiology and Immunology, Graduate School of Medicine, Hokkaido University, Kita 15, Nishi 7, Kita-ku, Sapporo 060-8638, Japan; ktakashima@med.hokudai.ac.jp (K.T.); oshiumi@kumamoto-u.ac.jp (H.O.); matumoto@pop.med.hokudai.ac.jp (M.M.); 2Department of Human Immunology, Research Institute for Frontier Medicine, Sapporo Medical University School of Medicine, South 1, West 17, Chuo-ku, Sapporo 060-8556, Japan; 3Department of Immunology, Graduate School of Medical Sciences, Kumamoto University, 1-1-1, Honjyo, Chuou-ku, Kumamoto 860-0811, Japan; 4Department of Pathology, National Institute of Infectious Diseases, 1-23-1, Toyama, Shinjuku, Tokyo 162-8640, Japan; ainai@nih.go.jp (A.A.); tksuzuki@niid.go.jp (T.S.); hasegawa@nih.go.jp (H.H.)

**Keywords:** IgA, NALT, cGAMP, STING

## Abstract

Induction of immunoglobulin (Ig) A in the mucosa of the upper respiratory tract and the nasal cavity protects against influenza virus infection. Cyclic dinucleotides (CDNs) are used as mucosal adjuvants to enhance the immunogenicity of intranasal influenza hemagglutinin (HA) vaccines. The adjuvant activity of 2′3′ cyclic guanosine monophosphate–adenosine monophosphate (cGAMP) on Ig production was investigated in nasal-associated lymphoid tissue (NALT), serum of wild-type C57BL/6J, and stimulator of interferon genes (STING)-deficient mice, which do not recognize cGAMP. Mice were vaccinated intranasally with a HA vaccine with or without the cGAMP adjuvant. IgA and IgG production, T-cell responses, germinal center formation, and cytokine expression in NALT were assayed. cGAMP enhanced IgA and IgG production, and promoted T-cell responses. Intranasal administration of cGAMP activated both NALT and systemic immune cells, induced a favorable cytokine environment for IgA induction, and promoted germinal center formation. The cGAMP effect was STING-dependent. Taken together, cGAMP as an HA vaccine adjuvant promoted a STING-dependent NALT environment suitable for the enhancement of IgA production.

## 1. Introduction

The influenza virus belongs to the family Orthomyxoviridae, and is a negative-sense single-stranded RNA virus. It causes annual worldwide epidemics of influenza disease [1]. Immunoglobulin (Ig) A has cross-reactivity with the virus, and IgA produced in the nasal and respiratory mucosa, where the influenza virus first encounters host cells, can block its entry [2]. It is required to develop effective vaccines and adjuvants which strongly induce IgA in the nasal and respiratory mucosa to prevent epidemics of influenza virus. The appropriate mucosal adjuvants activate the innate immune system and induce immune memory without side effects. The intranasal (i.n.) administration of hemagglutinin (HA) vaccines, in combination with pathogen-associated molecular pattern molecules (PAMPs) such as flagellin or polyinosinic-polycytidylic acid (polyI:C), or cyclic guanosine monophosphate (GMP)–adenosine monophosphate (AMP) (cGAMP), increase IgA production in the nasal cavity [3,4,5,6]. PAMPs are known as being mucosal adjuvants which activate innate immunity, but the mechanisms by which they stimulate antibody production are not well understood.

To develop novel adjuvants for i.n. vaccines, it helps to consider the mechanisms underlying the immune responses induced by known respiratory mucosal adjuvants. Epithelial and immune cells located in the oral and nasal mucosa are the first to respond to i.n. administration of viral antigens and adjuvants. Nasal-associated lymphoid tissue (NALT) located in the palate of the upper jaw contains T-cells, B-cells, dendritic cells (DCs), and macrophages that provide the initial response to i.n. vaccination [7]. We previously showed that polyI:C, a synthetic analog of double-stranded RNA, strongly induced HA-specific IgA production by CD103^+^ DCs in NALT after i.n. vaccination. The response was toll-like receptor (TLR) 3-dependent [8]. However, it is unclear how other mucosal adjuvants affect immune cells in NALT.

cGAMP is synthesized from cytosolic DNA by cGAMP synthase (cGAS) in mammalian cells and then binds to stimulator of interferon (IFN) genes (STING), inducing IFN regulatory factor 3-dependent and nuclear factor-kappa B-dependent production of type I IFN [9,10]. Cyclic dinucleotides (CDNs), including cGAMP, c-di-GMP, and c-di-AMP, act as mucosal adjuvants, enhancing Ig production and T-cell responses by binding to STING [6,11]. STING-mediated production of tumor necrosis factor (TNF)-α following i.n. administration of c-di-GMP augments type I IFN-independent production of antigen-specific Ig [11], and it also promotes antigen uptake in the lungs [6]. However, what the effects of cGAMP on early humoral immune responses in NALT after i.n. vaccination remain to be determined. To clarify the effects of cGAMP as a mucosal adjuvant, we focused on early immune responses involving cytokine expression, cell activation, and germinal center (GC) formation in NALT after intranasal inoculation with both vaccine and adjuvant in this study. Following i.n. vaccination, cGAMP strongly promoted STING-dependent activation of immune cells and formation of GCs. Enhancement of Ig production by cGAMP was not abolished in type I IFN receptor A (IFNAR)-defective mice, indicating that the adjuvanticity of cGAMP is independent of the IFN pathway.

## 2. Materials and Methods

### 2.1. Reagents

2′3′-cGAMP (cyclic [G(2′,5′)pA(3′,5′)p]) and polyI:C were purchased from InvivoGen ASIA (Shatin, Hong Kong) and GE healthcare Japan (Tokyo, Japan), respectively. The antibodies used in this paper are listed in Appendix A.

### 2.2. Mice

Wild-type (WT) C57BL/6J mice were purchased from CLEA Japan, Inc. (Tokyo, Japan). *Tmem173*^−/−^ mice were generated using the CRISPR/Cas9 system [12,13]. Knock-out mice were backcrossed with C57B6/J mice more than eight times before use. Age- and sex-matched WT and knockout mice were used for the experiments in this study. All mice were maintained under specific pathogen-free conditions in the Animal Facility at the Hokkaido University Graduate School of Medicine. The protocol was approved by the Committee for the Ethics of Animal Experiments in the Animal Safety Center, Hokkaido University, Japan. All mice were used according to the guidelines of the Institutional Animal Care and Use Committee of Hokkaido University, who approved this study as No. 16-0030 and 17-0015.

### 2.3. Vaccination and ELISA

Mice were anesthetized with pentobarbital sodium and xylazine and i.n. immunized by dropping 5 µL Phosphate balanced saline (PBS) containing 1 µg HA vaccine with or without 3 µg polyI:C or 2 µg cGAMP (InvivoGen, San Diego, CA, USA) as liquid drops into each nostril. Four weeks later, mice were re-immunized by the same method. Two weeks after the second immunization, blood samples and nasal wash (NW) fluid were collected from mice anesthetized by pentobarbital sodium and xylazine to measure vaccine-specific IgA and IgG antibody (Ab) production. The levels of IgA and IgG Abs against HA were measured by enzyme-linked immunosorbent assay (ELISA) as described previously, using plates coated with HA protein purified from the vaccine strain [3]. HA-coated plates were blocked with 1% bovine serum albumin /PBS for 2 h and the blocking buffer was discarded. Serum and NW were overlaid and then biotin-labelled goat anti-mouse IgG (Jackson ImmunoResearch, West Grove, PA, USA) or IgA (Kirkegaard & Perry Laboratories, Gaithersburg, MD, USA) were added. Streptavidin-alkaline phosphatase (Thermo Fisher Scientific, Waltham, MA, USA) was reacted with biotin and detection was performed using p-nitrophenyl phosphate (Sigma-Aldrich, St. Louis, MO, USA). The chromogen was measured by detection of absorbance at 405 nm using a SUNRISE plate reader (TECAN, Mannedorf, Switzerland). For preparation of standard IgG in serum and IgA in NW, Balb/c mice were immunized as described above. To determine ELISA units in standard samples, standard IgG in serum and IgA in NW were diluted in two-fold steps with blocking buffer, from 1:200 to 1:6,553,600 for serum and from 1:1 to 1:32,768 for NW, and ELISA was performed as described above. The standard IgG in serum and IgA in NW had ELISA endpoint titers of 1:409,600 and 1:2048, respectively. These endpoint titers of each standard for IgG in serum and IgA in NW were arbitrarily decided as being 1 ELISA unit. Standard IgG in serum and IgA in NW were incorporated into each ELISA assay to make a standard curve. The IgG and IgA ELISA units in test samples were calculated from their optical density. at 405 nm using the standard curve. Test serum samples and NW samples were diluted to 1:20 and 1:1, respectively.

### 2.4. Cytometric Bead Array Assay

Levels of cytokines produced in culture supernatants were measured by cytometric bead array (CBA) assay (BD Biosciences, Franklin Lakes, NJ, USA) following the manufacturer’s protocol. Prepared samples were analyzed using BD FACSAria II (BD Biosciences) and concentrations of cytokines were calculated by FCAP Array v3.0.1 (BD Biosciences).

### 2.5. Flow Cytometric Analysis

Spleens, lymph nodes (LNs), and NALT were collected from mice, and then tissues were mechanically mashed by slide glasses. Splenocytes, including red blood cells, were treated with red blood cell lysis buffer for 1 min, and then medium was added and collected by centrifuges. Cells from LNs and NALT were collected by centrifuges and subjected to the following assay. Cells prepared from NALT, LNs, and spleens were blocked with anti-CD16/CD32 Ab (93) and stained with the following fluorescence-labeled Abs; anti-B220 (RA3-6B2), anti-CD45 (30-F11), anti-CD19 (MB19-1), anti-IgA (mA-6E1), anti-CD4 (GK1.5), anti-CD8α (53-6.7), anti-CD11c (N418), anti-CD11b (M1/70), anti-CD80 (16-10A1), anti-CD86 (GL1), anti-CD69 (H1.2F3), anti-CD3 (145-2C11), anti-programmed cell death (PD)-1 (RMP1-30), anti-C-X-C chemokine receptor (CXCR)5 (L138D7), anti-Fas (Jo2), anti-GL7 (GL7), and anti-B cell activating factor (BAFF) (121,808). For intracellular staining, surface markers were stained and cells were fixed and permeabilized using a BD Cytofix/Cytoperm Fixation/Permeabilization Solution Kit (BD Biosciences) and then staining with anti-IFN-γ (XMG1.2). Abs were purchased from BioLegend (San Diego, CA, USA) and BD Biosciences. Stained cells were subjected to flow cytometry using Calibur and Aria II (BD Biosciences).

### 2.6. Antigen-Specific T-Cell Responses

Mice were vaccinated with either the vaccine or the vaccine and cGAMP twice. Two weeks after the second vaccination, splenocytes were collected from vaccinated mice and 1 × 10^6^ splenocytes were cultured in 96-well round bottom plates with 0–10 µg/mL of split-HA vaccine for 3 days. Cells were cultured with 10 µg/mL of brefeldin A (Sigma-Aldrich) durin g the last 6 h of culturing to inhibit protein transport. Cells were stained with fluorescein isothiocyanate (FITC)-conjugated anti-CD4, phycoerythrin (PE)-conjugated anti-CD8α, and PE-cyanine (Cy) 7-conjugated anti-CD3 Abs, and then fixed, permeabilized, and stained with allophycocyanin (APC)-conjugated anti-IFN-γ Ab. Prepared samples were subjected to flow cytometric analysis. The IFN-γ production in the culture supernatants was quantified with CBA assay (BD Biosciences) following the manufacturer’s instructions.

### 2.7. Real-Time Polymerase Chain Reaction

Total RNA was purified from NALT using TRIzol (Invitrogen, Carlsbad, CA, USA) following the manufacturer’s instructions. Total RNA was subjected to DNase I treatment (Takara BIO INC, Shiga, Japan). Real-time reverse transcription polymerase chain reaction (RT-PCR) was performed by a High Capacity complementary DNA (cDNA) Reverse Transcription kit (Applied Biosystems, City of Forster City, CA, USA) according to the manufacturer’s instructions. Real-time PCR was performed using a Step One real-time PCR system (Applied Biosystems). Primers used in this study are listed in Appendix A. Levels of target messenger ribonucleic acids (mRNAs) were normalized to β-actin and relative expression of transcripts was calculated using the double delta threshold cycles method relative to unstimulated samples.

### 2.8. Statistical Analyses

The statistical significance of differences between groups was determined by the Mann-Whitney U test or Kruskal-Wallis test with Dunn’s Multiple Comparison test using Prism 4 (GraphPad Software, San Diego, CA, USA). Values of *p* < 0.05 were considered significant.

## 3. Results

### 3.1. cGAMP Enhances Vaccine-Specific Ig Production

To evaluate the effect of cGAMP on vaccine-specific Ig production, WT mice were intranasally vaccinated with vaccines with or without cGAMP or polyI:C twice, with a 4-week interval between vaccines. PolyI:C was used as a positive adjuvant control. Serum and NWs were collected 2 weeks after the second vaccination for ELISA of IgA and IgG. As shown in Figure 1A,B, both cGAMP and polyI:C significantly enhanced production of vaccine-specific IgA and IgG in WT mice compared with the non-adjuvanted vaccine. The adjuvant effects of cGAMP and polyI:C appeared almost comparable in antibody production. cGAMP adjuvantation increased the proportion of IgA^+^ B-cells, but did not increase the total number of B220^+^ B-cells in NALT (Figure 1C,D). The results show that cGAMP enhanced the production of vaccine-specific IgG in serum and IgA in NALT.

### 3.2. cGAMP Promotes Vaccine-Specific T-Cell Responses

To evaluate the effect of cGAMP on vaccine-specific T-cell responses, we measured IFN-γ production in splenic T-cell preparations from vaccinated mice by flow cytometry. The proportions of IFN-γ^+^CD4^+^ and IFN-γ^+^CD8^+^ T splenocytes were significantly high in mice vaccinated with cGAMP compared with the non-adjuvanted vaccine (Figure 2A,B). Following restimulation with HA vaccine, IFN-γ, interleukin (IL)-6, IL-10, IL-17, and TNF-α concentrations in spleen cell supernatants were higher in preparations from mice given the adjuvanted vaccine than the non-adjuvanted vaccine (Figure 2C). cGAMP appeared to more efficiently produce cytokines in splenocytes than polyI:C in culture. The pattern of cytokine production in the spleens of mice vaccinated with cGAMP was like that observed in mice vaccinated with polyI:C (Figure 2C). IL-4 and IL-12p40 production was not detected in the culture supernatants (data not shown). The results suggest that cGAMP and polyI:C induce similar antigen-specific T-cell responses.

### 3.3. cGAMP Adjuvanticity is STING-Dependent

cGAMP binds to STING, resulting in production of inflammatory cytokines and type I IFNs [14]. To determine whether the adjuvant effects of cGAMP were STING- or IFNAR-dependent, *Tmem173*^−/−^ or *Ifnar*^−/−^ mice, where the STING protein or type I IFN receptor is deficient, respectively, were vaccinated with HA and cGAMP. cGAMP-enhanced IgA and IgG production was impaired in *Tmem173*^−/−^ but not *Ifnar*^−/−^ mice (Figure 3A,B). The proportion of IgA^+^ B cells in NALT increased in WT mice but not in *Tmem173*^−/−^ mice (Figure 3C,D). The results indicate that the adjuvant effect of cGAMP on Ig production was exerted through STING and that type I IFN was not required.

Cytokine production by splenocytes following restimulation with vaccine was assayed to evaluate the effects of STING deletion on T-cell responses. IFN-γ production in CD4^+^ and CD8^+^ T-cells following vaccination was significantly lower in *Tmem173*^−/−^ mice than in WT mice (Figure 4A,B). Pro-inflammatory and T-helper (Th) 1-, Th2-, and Th17-related cytokine concentrations were measured in culture supernatants using CBA assays. cGAMP increased IFN-γ, IL-17, IL-10, and TNF-α, but not IL-2 and IL-6 production in splenocytes from WT mice. The concentrations of all assayed cytokines were significantly lower in splenocyte preparations from *Tmem173*^−/−^ mice than in those from WT mice. The results show that the enhanced T-cell responses and cytokine production induced by cGAMP treatment were STING-dependent.

The post vaccination induction of germinal center (GC) B-cells and T follicular helper (Tfh) cells in NALT was investigated because the formation of the GC is required for class switching in vivo. The expression of CXCR5/PD-1 on CD4^+^ T-cells and of GL7/FAS on B-cells are known as Tfh and GC B-cell surface makers, respectively [15]. cGAMP significantly increased the proportions of Tfh and GC B-cells in NALT from WT but not *Tmem173*^−/−^ mice (Figure 4D,E); only marginal responses to cGAMP remained in *Tmem173*^−/−^ mice. The results indicate that cGAMP augmented STING-dependent T-cell responses and GC formation.

### 3.4. cGAMP Activates Immune Cells and Cytokine Expression in NALT via the STING Pathway

The activation of immune cells after vaccination with and without cGAMP were compared to clarify the mechanism of cGAMP enhancement of Ig production. cGAMP increased B-cell and DC activation in NALT, but did not increase the activation of CD3^+^ T-cells (Figure 5A and data not shown). T-cells, B-cells, and DCs were activated by cGAMP in cervical lymph nodes (CLN) (Figure 5B). cGAMP caused slight increases in splenic T- and B-cell activation; DC activation was not affected (Figure 5C). DCs were strongly affected in NALT; the activation of DCs and macrophages in CLNs and the spleen was less influenced by cGAMP (Figure 5A–C). There were no significant differences in activation states between WT and *Tmem173*^−/−^ mice in response to the non-adjuvanted vaccine (Figure 5A–C). The activation of immune cells was nearly abolished in *Tmem173*^−/−^ mice (Figure 5A–C). These results indicate that cGAMP induced activation of immune cells by the STING pathway, not only locally in NALT, but also systemically.

Changes in cytokine expression in NALT after i.n. vaccination with and without the cGAMP are shown in Figure 6. IFN-β mRNA expression was induced, and expression of IL-6 and IL-10, which are involved in IgA production, was upregulated in WT but not *Tmem173*^−/−^ mice. cGAMP also enhanced the expression of Il-1β and TNF-α, but not Il12p40 and Il-4 mRNA. Transforming growth factors (TGF)-β1–3, inducers of IgA class-switching, were unchanged by cGAMP, but IL-33 and thymic stromal lymphopoietin (TSLP), which are primarily produced by epithelial cells, were significantly enhanced in WT but not *Tmem173*^−/−^ mice by cGAMP inoculation (Figure 6). Although protein detection of these cytokines was difficult because of the limitations of harvested cell numbers, the results suggest that cGAMP promoted a cytokine environment in NALT in a STING-dependent manner, which resulted in the induction of IgA class switching.

## 4. Discussion

cGAMP was as effective as polyI:C in inducing IgA production and antigen-specific T-cell responses. The adjuvant activity was not type I IFN-dependent, but depended on STING recognition of cGAMP. cGAMP promoted GC development in NALT in a STING-dependent manner. IL-6 being detected in serum 2 h after intranasal inoculation of cGAMP (data not shown) suggest that part of cGAMP is circulated through the blood to attain DCs. According to our previous data [8], we speculate that CD103^+^ DCs located in NALT that respond to cGAMP migrate to CLNs and the spleen, resulting in the activation of immune cells, which induce the STING-dependent production of cytokines associated with IgA production in NALT. We showed for the first time that i.n. inoculation of cGAMP induces the activation of immune cells and the production of cytokines in NALT in early responses. The amounts of Ig production by cGAMP were comparable to those by polyI:C. The mice adjuvanted by polyI:C produced sufficient Ig to prevent death from attacks of the influenza virus [8]. These data suggest that cGAMP supplies enough Ig to protect mice against viral attacks when it is used as a mucosal adjuvant.

CDNs, including cGAMP and c-di-GMP, are known to function as mucosal adjuvants and to induce STING-dependent antigen-specific Ig production and T-cell responses [6,11,16,17,18]. Previous studies on *Tnfa*^−/−^ and TNF-α receptor (*Tnfr*)^−/−^ mice showed that the adjuvanticity of CDNs was type I IFN-independent and depended on STING-mediated TNF-α production [11]. HA-inhibiting titers are elevated in parallel with Ig levels in subjects with influenza vaccines in previous studies [19]. However, some reports found that GC formation and Ig production were reduced in *Tnfa*^−/−^ and *Tnfr*^−/−^ mice compared with WT mice [20,21,22]. Impaired Ig production by *Tnfr*^−/−^ mice compared with WT mice in response to CDN-adjuvanted vaccines might be more closely related to the *Tnfr*^−/−^ phenotype itself than to a lack of adjuvanticity. As *Tmem173*^−/−^ mice have a normal Ig response to cholera toxin, which is a strong mucosal adjuvant [23], decreased Ig production in response to cGAMP in *Tmem173*^−/−^ mice may be a cGAMP-specific phenomenon and not a consequence of the absence of STING.

CDNs induce balanced Th1, Th2, and Th17 cytokine responses in vivo that are needed for the enhancement of mucosal immunity by an adjuvant [17,24,25]. This study confirmed that cGAMP induced Th1-, Th2-, and Th17-type cytokines in vivo. This study also suggests that STING keeps active homeostasis of immune balance by recognizing DNA of self-origin, including mitochondria and dead cells. cGAMP also upregulated IL-33 and TSLP expression in NALT. TSLP activates DC production of Th2-type cytokines [26], which may enhance the differentiation of antigen-specific IgA-producing B-cells [26]. TSLP also induces the expression of B-cell activating factor by DCs, which promotes class switching by B-cells [27,28]. TSLP and IL-33 are primarily produced by epithelial cells [29,30]. Since CD11c-mediated Ig production by CDN-adjuvanted vaccine is abrogated in *Tmem173*^−/−^ mice compared to WT [6], CD11c^+^ DCs are responsible for the STING-dependent adjuvanticity. However, Ig production in CD11c-specific *Tmem173*^−/−^ mice is not completely impaired [6], suggesting that cells other than DCs are also involved in the STING-dependent adjuvanticity. As a mucosal adjuvant, cGAMP may act not only on immune cells but also non-immune cells, and promoting antigen uptake by lung DCs may also contribute to the adjuvant effect of CDNs [6].

Adding polyI:C to an intranasal HA vaccine strongly increases HA-specific IgA production in the nasal cavity, leading to resistance to influenza virus infection [3,4,31]. PolyI:C is recognized by endosomal TLR3 and cytosolic melanoma differentiation-associated gene 5/retinoic acid-inducible gene-I, which uses adaptor proteins, TLR adaptor molecule 1 (TICAM-1), and mitochondrial antiviral signaling (MAVS), respectively, to transduce signals that trigger an antiviral response [32]. We previously showed the adjuvanticity of polyI:C was dependent on the TLR3-TICAM-1 pathway using *Ticam-1*^−/−^ and *Mavs*^−/−^ mice [8]. As TLR3 expression is restricted to myeloid cells like CD8α^+^ and CD103^+^ DCs [33], polyI:C adjuvant activity depends on the activation of the TLR3-TICAM-1 pathway in CD103^+^ DCs in NALT [8]. STING is a cGAMP receptor and is widely expressed in immune and non-immune cells, which accounts for the induction of a systemic immune response to intranasal vaccination. The differential expression patterns of these innate sensors may be the reason for the differences in cytokine expression in splenocytes between polyI:C- and cGAMP-vaccinated mice. Combinational use of polyI:C and cGAMP would be possible, but would certainly increase adverse effects due to cytokine toxicity. In this experiment, we have no data with varying concentrations of cGAMP, so we cannot rule out the possibility that the concentration of cGAMP affects cytokine production after restimulation. Synthetic CDNs strongly activate systemic immunity and are undergoing clinical trials: CDNs would be hopeful adjuvants. Clinical trials of TLR7/9 agonists are still going on and are overcoming difficulties. The TLR3-specific RNA agonist with lesser toxicity shows efficient adjuvanticity in inducing cytotoxic T-cells in vivo [34,35]. Since Ig production in part depends on TLRs [8] and cytoplasmic sensors as shown here, which supports adjuvant trials to make mucosal IgA production effective.

## 5. Conclusions

The mucosal adjuvant activity of cGAMP includes activation of immune cells, induction of cytokine expression, and promotion of GC formation in NALT, leading to strong, STING-dependent production of IgA.

## Figures and Tables

**Figure 1 medsci-05-00035-f001:**
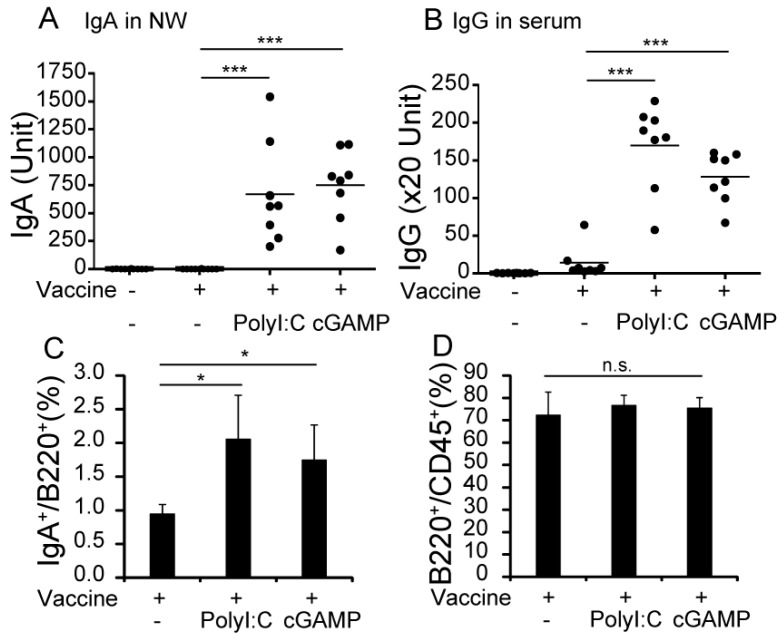
2′3′ cyclic guanosine monophosphate–adenosine monophosphate (cGAMP) enhances vaccine-specific Ig production in mice. Wild-type (WT) mice were intranasal (i.n.) vaccinated with 1 µg of hemagglutinin (HA) split vaccine with or without 2 µg of cGAMP or 3 µg of polyinosinic-polycytidylic acid (polyI:C) twice. Two weeks after the second vaccination, nasal wash (NW) and serum were collected to measure vaccine-specific immunoglobulin (Ig) A (**A**) and IgG (**B**) antibody (Ab) production by enzyme-linked immunosorbent assay (ELISA). The horizontal bars show the average of 8–10 mice for groups pooled from two independent experiments. Nasal-associated lymphoid tissue (NALT) cells were isolated from vaccinated mice and stained with fluorescein isothiocyanate (FITC)-labeled anti-IgA, phycoerythrin (PE)-labeled anti-B220, and allophycocyanin (APC)-labeled anti-CD45 Abs, and subjected to flow cytometric analysis. The proportions and numbers of IgA^+^ B cells and B220^+^ B cells in NALT were analyzed by flow cytometry. (**C**,**D**) The proportions of IgA^+^ in B220^+^ B cells and B220^+^ cells in CD45^+^ cells in NALT after vaccination. The bars show average of the 4–5 mice for each group. * *p* < 0.05; *** *p* < 0.001, n.s.: not significant in the Mann-Whitney U test. One representative experiment out of three is shown.

**Figure 2 medsci-05-00035-f002:**
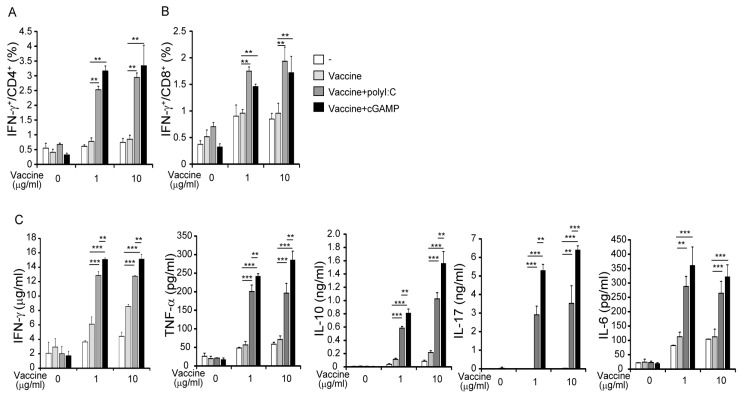
cGAMP-enhanced antigen-specific T-cell responses. WT mice were i.n. vaccinated twice and splenocytes (1 × 10^6^ cells) were re-stimulated by 0–10 µg/mL of split-HA vaccine for 3 days. Cells were incubated with brefeldin A during the last 6 h of culture and stained with PE-Cy7-labeled anti-CD3, FITC-labeled anti-CD4, PE-labeled anti-CD8 and APC-labeled anti-interferon (IFN)-γ Abs. The percentages of IFN-γ-producing CD3^+^CD4^+^ T-cells and CD3^+^CD8^+^ T-cells are shown in (**A**,**B**). (**C**) The levels of IFN-γ, interleukin (IL)-6, IL-10, IL-17, and tumor necrosis factor (TNF)-α production in culture supernatants were determined by cytometric bead array (CBA) assay. The values are presented as the mean ± standard deviation (SD) of 3 samples for each group. ** *p* < 0.01; *** *p* < 0.001 in the Mann-Whitney U test. One representative experiment out of three is shown.

**Figure 3 medsci-05-00035-f003:**
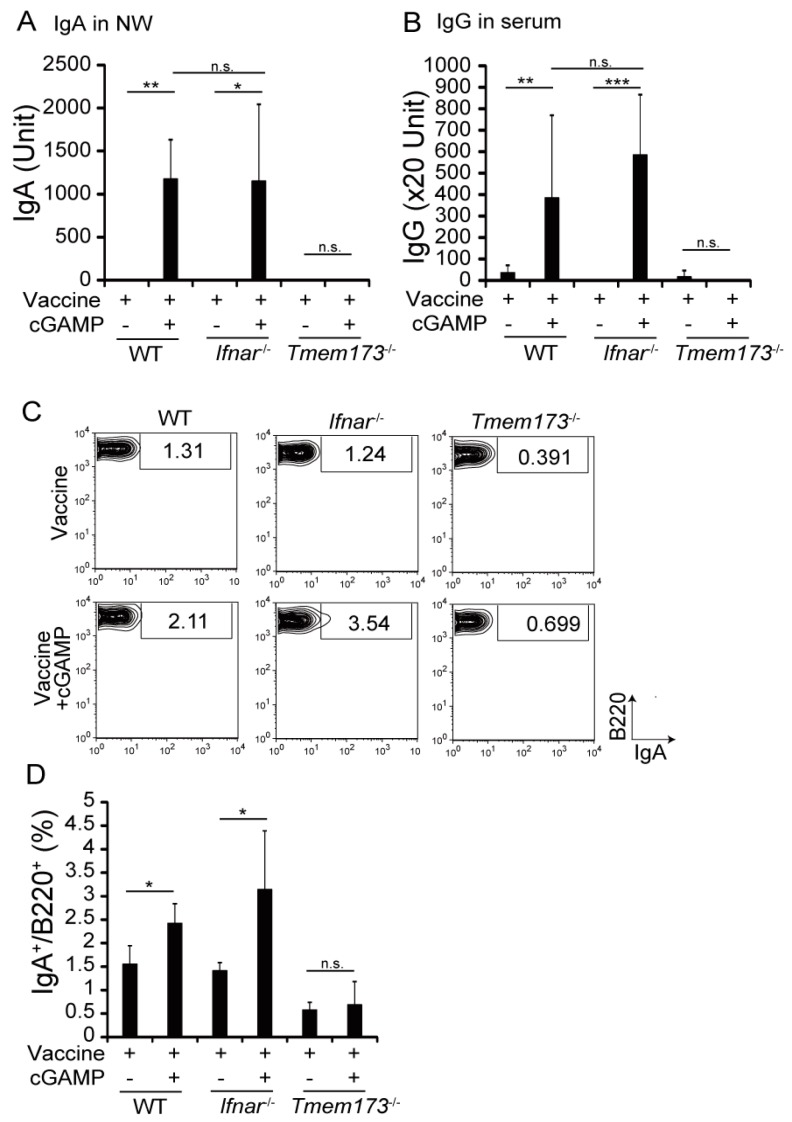
cGAMP enhanced Ig production in a stimulator of interferon genes (STING)-dependent manner. (**A**,**B**) NW and serum were collected from WT and indicated knockout mice vaccinated with the vaccine alone or the vaccine + cGAMP twice to measure HA-specific IgA and IgG by ELISA. The horizontal bars show the average of 8–10 mice for groups pooled from two independent experiments. n.s.: not significant; * *p* < 0.05; ** *p* < 0.01; *** *p* < 0.001 in the Kruskal-Wallis test with Dunn’s Multiple Comparison test. (**C**,**D**) Cells were isolated from NALT after vaccination, and stained with FITC-labeled anti-IgA, PE-labeled anti-B220, and APC-labeled anti-CD45 Abs. (**C**) One representative experiment out of three is shown. The indicated numbers show the percentages of gated population. (**D**) The graph shows percentages of IgA^+^/B220^+^ cells in NALT. The values are presented as the mean ± SD of 4 samples for each group. n.s.: not significant; * *p* < 0.05; ** *p* < 0.01; *** *p* < 0.001 in the Mann-Whitney U test.

**Figure 4 medsci-05-00035-f004:**
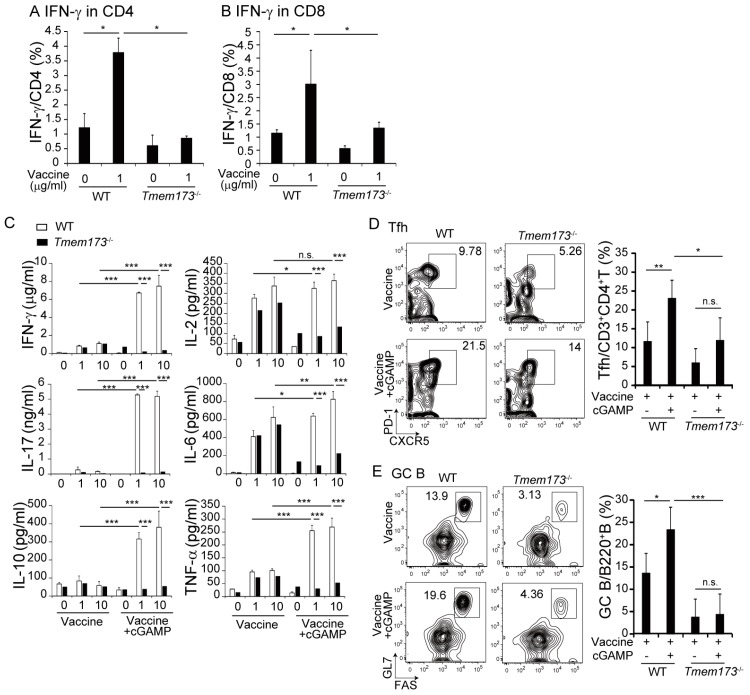
cGAMP augments cytokine production and T-cell responses in a STING-dependent manner. WT mice were i.n. vaccinated twice and splenocytes (1 × 10^6^ cells) were re-stimulated by 0–10 µg/mL of split-HA vaccine for 3 days. Cells were incubated with brefeldin A during the last 6 h of culturing and stained with PE-Cy7-labeled anti-CD3, FITC-labeled anti-CD4, PE-labeled anti-CD8, and APC-labeled anti-IFN-γ Abs. The percentages of IFN-γ-producing CD3^+^CD4^+^ T-cells and CD3^+^CD8^+^ T-cells are shown in (**A**,**B**). (**C**) The levels of IFN-γ, IL-2, IL-6, IL-10, IL-17, and TNF-α production in culture supernatants were determined by CBA assay. The values are presented as the mean ± SD of 3 samples for each group. (**D**) Flow cytometry profiles of NALT germinal center (GC) B-cells. GC B-cells were stained with APC-labeled anti-CD45, PE-Cy7-labeled anti-Fas, peridinin-chlorophyll protein (PerCP) Cy5.5-labeled anti-GL7, and FITC-labeled anti-B220 Abs 2 weeks after vaccination. One representative experiment out of four is shown. The graph shows the percentage of GC B cells. (**E**) Flow cytometry of NALT T follicular helper (Tfh) cells stained for APC-labeled anti-programmed cell death (anti-PD)-1, PE-Cy7-labeled anti-CD3, PerCP-Cy5.5-labeled anti-CXCR5, Alexa700-labeled anti-CD45, and FITC-labeled anti-CD4 Abs 2 weeks after vaccination. One representative experiment out of four is shown. The graph shows the percentage of Tfh cells. The values are presented as the mean ± SD of 4 mice for each group. * *p* < 0.05; ** *p* < 0.01; *** *p* < 0.001 and n.s.: not significant in the Mann-Whitney U test. The indicated numbers show the percentages of gated populations.

**Figure 5 medsci-05-00035-f005:**
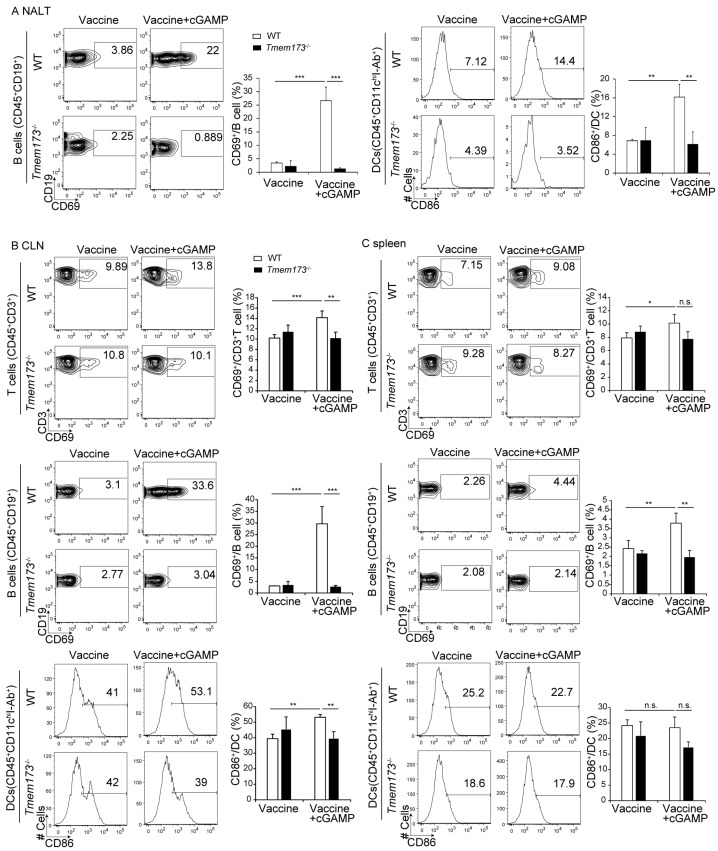
cGAMP activates immune cells in NALT. Cells were isolated from NALT, cervical lymph nodes (CLN), and spleens, which were harvested from WT mice vaccinated with the vaccine alone or the vaccine + cGAMP for 24 hrs. (**A**) (Left panel) NALT cells were stained with 7AAD, PE-labeled anti-CD69, Alexa Fluor 700-labeled anti-CD19, and APC-Cy7-labeled anti-CD45 Abs for B-cells. Cells gated with 7 aminoactinomycin D (AAD)^−^CD45^+^CD19^+^ cells are shown as B-cells. (Right panel) NALT cells were stained with 7AAD, FITC-labeled anti-CD86, PE-labeled anti-I-Ab, PECy7-lebeled anti-CD11c Abs, and APCCy7-labeled anti-CD45 Abs for dendritic cells (DCs). Cells gated with 7AAD^−^CD45^+^CD11c^hi^I-Ab^+^ cells are shown as DCs. (**B**,**C**) Flow cytometry analysis of cells prepared from CLNs (**B**) and spleens (**C**), stained with Abs by the same protocol as shown in (**A**). Cells stained with 7AAD, FITC-labeled anti-CD69, PECy7-labeled anti-CD3, and APCCy7-labled anti-CD45 Abs. Cells gated with 7AAD^−^CD45^+^CD3^+^ are shown as T-cells. One representative experiment out of three is shown. The indicated numbers show the percentages of gated populations. The values are presented as the mean ± SD of 3 mice for each group. * *p* < 0.05; ** *p* < 0.01; *** *p* < 0.001 and n.s.: not significant in the Mann-Whitney U test.

**Figure 6 medsci-05-00035-f006:**
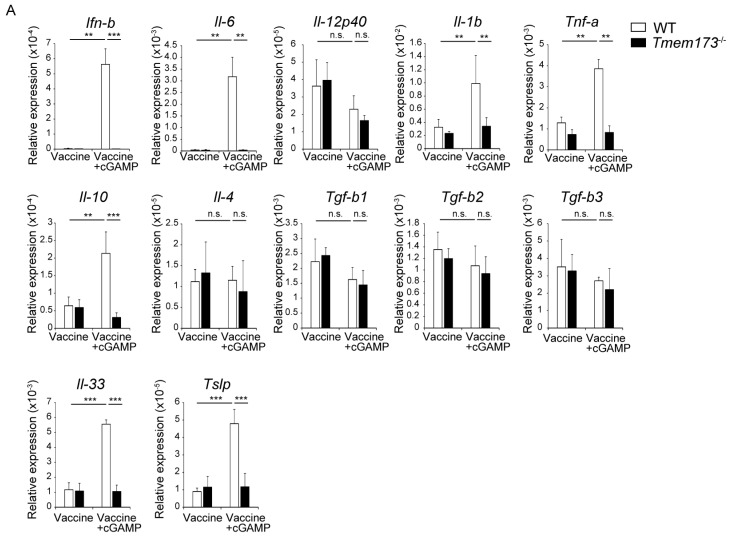
Messenger ribonucleic acid (mRNA) expression in NALT after i.n. vaccination with cGAMP. Levels of the indicated mRNA expression in NALT prepared from mice vaccinated with either vaccine alone or vaccine + cGAMP for 6 h were determined by real-time PCR. Levels of target mRNAs were normalized to β-actin and graphs show relative expression of target mRNAs. The values are presented as the mean ± SD of 3 mice for each group. ** *p* < 0.01; *** *p* < 0.001 and n.s.: not significant in the Mann-Whitney U test.

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
