# Peer review of "cGAMP Promotes Germinal Center Formation and Production of IgA in Nasal-Associated Lymphoid Tissue"

_medsci, 2017, doi:10.3390/medsci5040035_

Reviewer 1 Report

In the manuscript "cGAMP promotes germinal center formation and the production of IgA in nasal-associated lymphoid tissue" by Takaki H et al., the authors examined the mucosal adjuvant activity of cGAMP in vivo. They focused on the cGAMP response in NALT and found that cGAMP adjuvant activity depends on TMEM173 but not type I IFN. They suggested that cGAMP induces germinal center formation in NALT that promotes IgA production. Overall the research is sound. I have some minor points below.

Some minor points:

In line 161, "cGAMP adjuvantation increased the proportion of IgA+ B-cell... (Figure 1C) ". There were no statistics in Figure 1C to indicate the difference is significant.

In line 233, the author stated "cGAMP significantly increased the proportions of Tfh and GC B cells in NALT from WT but not Tmem173-/- mice". However,  in Figure 4D, Tfh in Tmem173-/- samples also increased from 5.26% to 14%. The author may want to do intracellular Bcl6 stain, alone with CCR5 and PD-1. 

In line 303, "cGAMP activated immune cells in NALT, CLNs and the spleen", did the authors imply that intranasal administered cGAMP (in 5ul volume) trafficked to CLN, spleen and directly activate immune cells there or cGAMP activated immune cells in NALT, which trafficked to CLN, spleen? 

In line 342, "As cGAMP strongly activates systemic immunity, it may not be an ideal adjuvant for human vaccines because systemic activation of B-cells could induce production of autoantibodies and autoimmune disorders." This statement is rather odd. First, synthetic CDNs, which is much more potent than cGAMP, are currently in clinical trial. So it is premature to say that cGAMP would not be an ideal human vaccine. Second,  did the authors see the development of autoimmune diseases in their cGAMP immunized mice that warranted such an argument?

Author Response

Reviewer #1

In the manuscript "cGAMP promotes germinal center formation and the production of IgA in nasal-associated lymphoid tissue" by Takaki H et al., the authors examined the mucosal adjuvant activity of cGAMP in vivo. They focused on the cGAMP response in NALT and found that cGAMP adjuvant activity depends on TMEM173 but not type I IFN. They suggested that cGAMP induces germinal center formation in NALT that promotes IgA production. Overall the research is sound. I have some minor points below.

Thank you for your comments to improve our manuscript.

Some minor points:

In line 161, "cGAMP adjuvantation increased the proportion of IgA+ B-cell... (Figure 1C) ". There were no statistics in Figure 1C to indicate the difference is significant.

As reviewer pointed, we added the results of statistical analysis in Figure 1c.

In line 233, the author stated "cGAMP significantly increased the proportions of Tfh and GC B cells in NALT from WT but not Tmem173-/- mice". However, in Figure 4D, Tfh in Tmem173-/- samples also increased from 5.26% to 14%. The author may want to do intracellular Bcl6 stain, alone with CCR5 and PD-1.

Although we performed a significant difference test between vaccine-treated Tmem173-/- mice vs vaccine+cGAMP-treated Tmem173-/- mice, no significant difference was detected in both groups (p = 0.422). Therefore, we concluded that cGAMP treatment exert no effect on Tfh proportion in Tmem173-/- mice. We added the “n.s.” in Figure 4D.

We also mentioned the marginal response to cGAMP in NALT from Tmem173-/- mice" from the text (line 239 in the new text).

In line 303, "cGAMP activated immune cells in NALT, CLNs and the spleen", did the authors imply that intranasal administered cGAMP (in 5ul volume) trafficked to CLN, spleen and directly activate immune cells there or cGAMP activated immune cells in NALT, which trafficked to CLN, spleen?

The reviewer pointed our over-interpretation and we found the comment right. We showed no experimental evidence regarding the trafficking of cGAMP. Thus, we modified the sentence in line 308-312 of the new text.

In line 342, "As cGAMP strongly activates systemic immunity, it may not be an ideal adjuvant for human vaccines because systemic activation of B-cells could induce production of autoantibodies and autoimmune disorders." This statement is rather odd. First, synthetic CDNs, which is much more potent than cGAMP, are currently in clinical trial. So it is premature to say that cGAMP would not be an ideal human vaccine. Second, did the authors see the development of autoimmune diseases in their cGAMP immunized mice that warranted such an argument?

We agree to the reviewer's view, and soften the sentence. We should have mentioned that clinical trials of inflammatory adjuvants including TLR7/9 agonists were dropped out. We are aware that CDNs are more potent than cGAMP and hopeful. We include this issue in the discussion (in line 358-362). We omitted the sentence of the autoimmune disorders. 

Reviewer 2 Report

Major Concerns:

The authors show that  in cGAMP vaccinated groups, humoral and cell-mediated responses in Tmem173-/- mice are impaired. However, compared to WT mice, Tmem173-/- mice also showed reduced percentages of IgA expressing B cells (Figure 3c and 3d) and reduced T cell response (Figure 4) in the absence of cGAMP. The authors should comment on these findings.  

2. In Figure 2c, the authors show that mice vaccinated with cGAMP adjuvanted vaccine produced significantly higher levels of IL-6 when compared to vaccine alone. However, in Figure 4c, although a similar experimental design was used, there is no observable increase in IL-6 production (cGAMP adjuvanted versus vaccine alone). The authors should comment on this discrepency.

Minor Concerns:

1. Criteria for a proper mucosal adjuvant should be clarified in introduction part. 

2. In materials and methods part, single cell preparation from NALT, LN and spleen can be explained more detailed.

3. Figure 1c and 1d lack statistical analysis.

4. Name of the STING knockout mice was written incorrectly multiple times (figure 3, 4, 5 and 6).  It is written as Tmem137-/- while it should be Tmem173-/-.

5. In figure 4c, results of statistical analysis was neglected.

6. In figure 5, which bar (black or white) corresponds to which mice (WT or Tmem137-/-) should be indicated.

7. Resolution and quality of figures should be enhanced, especially for figure 5.

Author Response

Reviewer#2

The authors show that in cGAMP vaccinated groups, humoral and cell-mediated responses in Tmem173-/- mice are impaired. However, compared to WT mice, Tmem173-/- mice also showed reduced percentages of IgA expressing B cells (Figure 3c and 3d) and reduced T cell response (Figure 4) in the absence of cGAMP. The authors should comment on these findings. 

Thank you for your useful indications. We noticed that IgA+B cells and T cell response in Tmem 173-/- mice were statistically lower than those in WT mice. STING, which is the protein coded by Tmem173, recognizes various self DNA including mitochondrial DNA and self DNA produced from apoptosis cells to keep homeostasis (Abe T et. al., Molecular Cell, 50, 5-15 (2013)). Therefore, we believe that there are possibility that immune response in steady state is decreased in Tmem173-/- mice. However, Tmem173 KO mice are able to produce normal Ig in response to cholera toxin, which is a major mucosal adjuvant, so these KO mice have normal immune system in response to mucosal adjuvants without cGAMP. The point is reflected in the text (line 337-340).

2. In Figure 2c, the authors show that mice vaccinated with cGAMP adjuvanted vaccine produced significantly higher levels of IL-6 when compared to vaccine alone. However, in Figure 4c, although a similar experimental design was used, there is no observable increase in IL-6 production (cGAMP adjuvanted versus vaccine alone). The authors should comment on this discrepency.

We thank the reviewer for this comment. As reviewer pointed, we performed significant difference tests in Figure 4. We observed significant difference between level of IL-6 in vaccine alone and vaccine+cGAMP-treated WT mice in Figure 4. We added the results of statistical analysis in Figrue 4.

Minor Concerns:

1. Criteria for a proper mucosal adjuvant should be clarified in introduction part.

As reviewer pointed, we added the sentence in line 43-44.

2. In materials and methods part, single cell preparation from NALT, LN and spleen can be explained more detailed.

As reviewer pointed, we added the sentence in line 123-126 in the new text.

3. Figure 1c and 1d lack statistical analysis.

As reviewer pointed, we added results of statistical analysis in Figure 1c and d.

4. Name of the STING knockout mice was written incorrectly multiple times (figure 3, 4, 5 and 6).  It is written as Tmem137-/- while it should be Tmem173-/-.

Thank you for your pointing out. We corrected Tmeme137 to Tmem173.

5. In figure 4c, results of statistical analysis was neglected.

As reviewer pointed, we added results of statistical analysis in Figure4c.

6. In figure 5, which bar (black or white) corresponds to which mice (WT or Tmem137-/-) should be indicated.

Thank you for your pointing out. We added the legend in Figure 5.

7. Resolution and quality of figures should be enhanced, especially for figure 5.

Thank you for your pointing out. We have replaced all Figures with high resolution.

Reviewer 3 Report

This manuscript is very wisely written with minimal descriptions of data. The authors followed a very standard way of showing adjuvant activities of PAMPs and employed appropriate animal models. Several additional data will make this work more impactive.

(1) Which cells are responsible for the STING-dependent adjuvanticity? It seems that innate immune cells in NALT or CLN should play crucial roles in stimulating adaptive immune cells in the secondary lymphoid organs. More plausible discussions should be provided with direct/indirect experimental evidence.

(2) For the protective effect of influenza vaccines, the HAI antibody response in the systemic compartment plays essential roles and serves surrogate marker of efficacy. Have the authors assayed HAI titers?

Reviewer 3 Report

This manuscript is very wisely written with minimal descriptions of data. The authors followed a very standard way of showing adjuvant activities of PAMPs and employed appropriate animal models. Several additional data will make this work more impactive.

(1) Which cells are responsible for the STING-dependent adjuvanticity? It seems that innate immune cells in NALT or CLN should play crucial roles in stimulating adaptive immune cells in the secondary lymphoid organs. More plausible discussions should be provided with direct/indirect experimental evidence.

(2) For the protective effect of influenza vaccines, the HAI antibody response in the systemic compartment plays essential roles and serves surrogate marker of efficacy. Have the authors assayed HAI titers?

Author Response

Reviewer#3

This manuscript is very wisely written with minimal descriptions of data. The authors followed a very standard way of showing adjuvant activities of PAMPs and employed appropriate animal models. Several additional data will make this work more impactive.

Thank you for your comments.

(1) Which cells are responsible for the STING-dependent adjuvanticity? It seems that innate immune cells in NALT or CLN should play crucial roles in stimulating adaptive immune cells in the secondary lymphoid organs. More plausible discussions should be provided with direct/indirect experimental evidence.

Thank you for your comments. We discussed this issues in line 337-340 in the new text.

(2) For the protective effect of influenza vaccines, the HAI antibody response in the systemic compartment plays essential roles and serves surrogate marker of efficacy. Have the authors assayed HAI titers?

Thank you for your comments. We added the discussion in line 312-315 and 321-322 (see the new text).
